# Wireless Communication Platform Based on an Embroidered Antenna-Sensor for Real-Time Breathing Detection

**DOI:** 10.3390/s22228667

**Published:** 2022-11-10

**Authors:** Mariam El Gharbi, Raúl Fernández-García, Ignacio Gil

**Affiliations:** Department of Electronic Engineering, Universitat Politècnica de Catalunya, 08222 Terrassa, Spain

**Keywords:** antenna sensor, breathing status, embroidered textile, e-textile, wearable system

## Abstract

Wearable technology has been getting more attention for monitoring vital signs in various medical fields, particularly in breathing monitoring. To monitor respiratory patterns, there is a current set of challenges related to the lack of user comfort, reliability, and rigidity of the systems, as well as challenges related to processing data. Therefore, the need to develop user-friendly and reliable wireless approaches to address these problems is required. In this paper, a novel, full, compact textile breathing sensor is investigated. Specifically, an embroidered meander dipole antenna sensor integrated into an e-textile T-shirt with a Bluetooth transmitter for real-time breathing monitoring was developed and tested. The proposed antenna-based sensor is designed to transmit data over wireless communication networks at 2.4 GHz and is made of a silver-coated nylon thread. The sensing mechanism of the proposed system is based on the detection of a received signal strength indicator (RSSI) transmitted wirelessly by the antenna-based sensor, which is found to be sensitive to stretch. The respiratory system is placed on the middle of the human chest; the area of the proposed system is 4.5 × 0.48 cm^2^, with 2.36 × 3.17 cm^2^ covered by the transmitter module. The respiratory signal is extracted from the variation of the RSSI signal emitted at 2.4 GHz from the detuned embroidered antenna-based sensor embedded into a commercial T-shirt and detected using a laptop. The experimental results demonstrated that breathing signals can be acquired wirelessly by the RSSI via Bluetooth. The RSSI range change was from −80 dBm to −72 dBm, −88 dBm to −79 dBm and −85 dBm to −80 dBm during inspiration and expiration for normal breathing, speaking and movement, respectively. We tested the feasibility assessment for breathing monitoring and we demonstrated experimentally that the standard wireless networks, which measure the RSSI signal via standard Bluetooth protocol, can be used to detect human respiratory status and patterns in real time.

## 1. Introduction

In recent years, the market for wearable technologies involving advanced electronic technologies, computer sciences and fashion has experienced significant growth. Among diverse wearable technologies, e-textiles are one of the most popular categories, with great influence in our daily lives and prevalence in many applications, including military, aerospace, automotive, and medical fields [1]. The e-textiles are based on adding sensing electronic components to clothes or creating fabrics based on conductive fibers/yarns/printing inks in order to sense and respond to various external and environmental effects: mechanical, electrical, chemical, etc. [2,3]. In the medical field in particular, e-textiles have been widely used for continuous monitoring of vital signs [4]. To acquire vital signs from a person, there are different types of devices, but in general, the measurement technique is divided into two categories: wired and wireless. Wired systems are generally attached to the human body to collect a vital sign using cables. These kinds of systems are often used in hospitals and medical centers to measure vital signs on a daily basis, although several patients face difficulties or discomfort during the measurement, particularly burn victims or child patients [5]. As a consequence, these techniques have restrictions for continuous monitoring, because any activity or movement is not accepted, and this limitation can bother the patient. In contrast, wireless systems are gaining more popularity because they allow long-term usability and provide convenient continuous monitoring of vital signs that could provide additional understanding about disease progression, allowing prompt clinical intervention because the data is sent and received wirelessly in real time [6,7].

Breath is one of the most important vital signs, and it is considered as a crucial indicator to evaluate the physiological, physical state of human health, including diseases such as sleep apnea, asthma and cardiopulmonary diseases [8,9]. Therefore, continuous monitoring of breathing plays an important role in assessing a person′s health status. The growth of demand for optimized medical breathing monitoring systems has led intensive research to obtain simple, comfortable and accurate measurement solutions [10]. In the literature, different methods were proposed for breathing monitoring. Breathing status assessment is usually evaluated through human body temperature, movements or sounds. Specifically, several techniques were proposed, such as the technique based on the modulation of cardiac activity, the technique based on air temperature and the technique based on chest wall movement analysis [11]. One of the techniques based on the modulation of cardiac activity is an electrocardiogram (ECG), which requires the distribution of electrodes on the patient’s body [12]. The main drawback of this technique is the complexity of the equipment and difficulty of implementation. Another technique widely used for breathing monitoring is the Respiratory Inductive Plethysmography (RIP) method [13]. This technique was developed to evaluate respiratory status by measuring the abdominal wall and movement of the chest. The disadvantage of this method is that the equipment is expensive and cumbersome. The drawback of these methods is the complexity of their implementation for daily use by the patient due to the wired connection. In addition, they require complex techniques for processing data [14].

Recently, there has been an increasing demand for wireless respiratory monitoring methods, as they are practical to use for convenient and long-term continuous monitoring of patients′ physiological status. These wireless techniques provide an excellent user experience because they usually involve a sensing system located at a remote location so that a person can perform their usual activities while they are continuously monitored. Various contactless methods were proposed to monitor breathing, such as ultra-wideband (UWB) radars [15], camera-based systems [16], thermal imaging [17] and infrared thermography based on wavelet decomposition [18]. Although these techniques are contactless, their main drawback is that they suffer from difficulty of usage and inaccuracy.

Recently, new noninvasive and contactless alternatives have emerged that rely on integrating sensors into clothing, which provides more comfortable and user-friendly approaches for breathing monitoring. Many studies have proposed sensing systems integrated into textiles for breathing monitoring, including fiber Bragg grating-based sensors [19], magnetometers [20], inertial measurement unit sensors [21] and multimaterial fiber antenna sensors [22]. Although the proposed sensor systems embedded in textiles were able to detect vital signs, methods for data processing and analysis must be developed, and user comfort is still a requirement.

In this work we investigate, design and test a textile sensor for real-time breathing detection. In particular, an embroidered meander dipole antenna-based sensor integrated into an e-textile T-shirt with a Bluetooth transmitter for real-time breathing monitoring is proposed in this work. The antenna-based sensor is designed to transmit data over wireless communication networks at 2.4 GHz. The breathing detection is based on the received signal strength indicator (RSSI) measurements. The respiratory system is placed on the middle of the human chest. During breathing, the transmitted signal from the antenna-based sensor is sensitive to the stretching caused by the expansion/contraction of the chest, so it can be used for monitoring the respiratory signal. The respiratory signal is extracted from the variation of the RSSI signal emitted at 2.4 GHz from the embroidered antenna-based sensor integrated into a commercially made T-shirt and detected using a base station (laptop). The paper is organized as follows: Section 2 presents the design of the proposed system and the approach for the sensing and data collection. Section 3 shows the test results of breathing status. The discussion based on the results can be found in Section 4. Finally, conclusions are drawn in Section 5.

## 2. Design and Fabrication of E-Textile T-Shirt for Breathing Monitoring

### 2.1. Mechanism of Breath Detection

During respiration, the volume of the breathing system changes due to the movement of the chest wall and abdomen caused by contractions of the diaphragm and the intercostal muscles. Figure 1 presents a schematic representation of the ventral body cavity. The contraction of the diaphragm leads the abdominal organs down, causing a decrease in intrathoracic pressure. There are two various sorts of physical movements during a respiratory period: breathing in, or inhalation (draws air into the lungs), and breathing out, or exhalation (pushes air out of the lungs). During inhalation, the intercostal muscles contract and the size of the thoracic cavity increases, and this is related to the lowering of the diaphragm. Thus, this reduces the pressure within the alveolus so that air flows into the lungs, as shown in Figure 1a. During exhalation, the intercostal muscles and diaphragm relax, the abdomen and chest return to a resting position and lung volumes decrease, as shown in Figure 1b.

The breathing sensing mechanism of the proposed system is based on the operation frequency shift of the embroidered antenna sensor. When the textile antenna-based sensor is worn by a human body, its geometry is subject to significant deformation due to the body’s respiration (expansion/contraction). As a result, the operation frequency of the embroidered antenna shifts downward or upward because of the strain applied on the antenna by the chest movement. The breathing antenna sensor is connected to a Bluetooth transmitter, which is placed on the middle of the chest position. The detuning of the antenna frequency generates a mismatching in the transmitter, which affects the received RSSI. A base station contains a laptop with a receiver Bluetooth module prepared to continuously record the variation of the RSSI signal emitted by the embroidered antenna sensor.

### 2.2. Embroidered Antenna-Based Sensor Design

The textile antenna-based sensor is a critical component of wearable technologies, especially in an e-textile system that implements wireless communication, localization and sensing functions while it is integrated unobtrusively and comfortably into the clothes. The implementation of the antenna-based sensor in textile materials first requires a dielectric material characterization, as shown in Figure 2a. In this particular case, a split-post dielectric resonator (SPDR) was used to determine the dielectric properties of the T-shirt by means of the resonance method. The substrate is the cotton of a commercially available T-shirt. The experimental dielectric constant and loss tangent of the cotton substrate were $\varepsilon_{r}\;$ = 1.3 and tanδ = 0.0058, respectively. Also, an Electronic Outside Micrometer was used to measure the thickness (h) of the T-shirt, obtaining h = 0.464 mm. After measuring all the necessary parameters of the substrate, the proposed antenna-based sensor was designed using the commercial CST Studio Suite 3D full electromagnetic simulator 2021. For the design process, we started with a conventional dipole antenna, which has a large size, and we carried out a parametric study of different parameters in order to minimize the size and to optimize its behavior at 2.4 GHz. Moreover, the overall size of the proposed meander dipole antenna was reduced by up to 54% compared to the normal textile dipole reported in the literature [23]. A meander dipole antenna-based sensor was designed for compactness. Its dimensions are presented in Figure 2b: w = 45 mm, L = 4.8 mm, d = 7.6 mm and g = 2 mm. The layout of the proposed model is converted to a stitch pattern by using Digitizer Ex software (Figure 2c). This software is used in order to obtain a digital stitch format file that can be read by the Singer Futura XL550 embroidery machine. The final prototype of the proposed antenna-based sensor is depicted in Figure 2d. A commercial Shiel-dex 117/17 dtex 2-ply was used for the conductive layer. This commercial yarn is made of 99% pure silver-plated nylon yarn 140/17 dtex, which provides excellent conductivity. In addition, this conductive yarn allows the antenna-based sensor to be integrated into the cotton T-shirt without mobility restriction or compromising the comfort of the person under test, due to its high flexibility. Figure 2 summarizes all the fabrication steps of the proposed embroidered antenna-based sensor.

### 2.3. Respiratory Antenna-Sensor Based on RSSI Measurement

A novel detection system based on the measurement of the power transmitted by an embroidered meander dipole antenna-based sensor through a Bluetooth protocol was presented. The proposed breathing system is composed of four parts: an embroidered meander dipole antenna-based sensor integrated into a cotton T-shirt and placed on the middle of the human chest, a transmission Bluetooth module, a receiver Bluetooth module and a base station. The detection base station could be a laptop or tablet with a receiver Bluetooth module. The breathing antenna sensor is connected to a Bluetooth transmitter, which is placed on the middle of the chest position, maximizing the user′s comfort with no restrictions of movement. The Bluetooth transmitter module is fabricated by means of a commercial ESP32-WROOM-32UE [18], which contains a U. FL connector that needs to be connected to an external antenna. In our case, the external antenna was a fully embroidered meander dipole antenna-based sensor integrated into a T-shirt, as described in the previous section. The fabricated module was stitched by conductive yarn into the T-shirt by means of two metallic pads to insure a good connection with the embroidered antenna sensor. The ESP32 used in the fabricated transmitter module is a low-cost, low-power system on a chip (SoC) series with Bluetooth and Wi-Fi capabilities. It is often used for wearable electronics, mobile devices, and IoT applications. This transmission module acts as an advertising beacon that follows the Bluetooth Low Energy (BLE) protocol, with an operating frequency of 2.4 GHz. The transmitter Bluetooth module was fabricated and sewn into the T-shirt. A commercial ESP32-WROOM-32U was used for programming our fabricated transmitter module. The block diagram for the proposed breathing monitoring system is presented in Figure 3. The custom board has a compact size 2.36 × 3.17 cm^2^, as shown in Figure 3, which provides a high comfort level for the user wearing the T-shirt. The proposed breathing wireless system was sewn into the T-shirt. To extract the received signal strength indicator through Bluetooth protocol, a code was developed using Arduino Software (IDE) to display the variation of the RSSI signal from the physical detuning of the antenna sensor embedded into the T-shirt during the respiration process. The received power of a BLE signal is disposable in the form of a received signal strength indicator (RSSI), which is given by Bluetooth. It is an integer value representing the received power in dBm. The breathing signal is extracted from the variation of the received signal strength indicator (RSSI) from the antenna sensor embedded into the T-shirt. The RSSI measurement was detected using a laptop connected to a receiver Bluetooth module (ESP32 ESP-WROOM-32). This receiver module is a low-cost and low-power system on a chip microcontroller with integrated dual-mode Bluetooth and Wi-Fi. It is well suited to a number of different IoT (Internet of Things) devices, such as smart security devices (smart locks and surveillance cameras), smart industrial devices and smart medical devices (wearable health monitors). The respiration signal is obtained from the RSSI signal transmitted by the antenna-based sensor, which is sensitive to the strain caused by the movement of the abdomen and chest wall. After programming the receiver Bluetooth module, the receiver module receives the Bluetooth signal from the transmitter antenna sensor, and it estimates the received power of a BLE signal through RSSI data and sends all the obtained values via a serial connection to the laptop. MATLAB was used to collect and visualize the breathing data information in real time.

## 3. Experimental Measurement and Performance Analysis

Breathing pattern monitoring is a mandatory and crucial task in healthcare fields. Indeed, the respiratory signal plays a key role in the estimation and prediction of the physiological status of patients. To acquire a breathing signal from a person, diverse systems and techniques are developed, but most of these techniques are generally attached to the human body using wires. This leads to difficulties and discomfort for the patients during the measurement, especially in burn victims or child patients. To improve user comfort, we proposed an e-textile T-shirt by means of a wireless system that was stitched to an embroidered antenna sensor for real-time breathing detection. During expiration (exhalation) and inspiration (inhalation), shape variation of the chest provides quantitative information about human breathing status. The transmitted signal from the embroidered antenna-based sensor is obtained and recorded by the movements of the chest and the abdominal wall. Note that the transmitted signal from the antenna sensor is sensitive to stretch caused by the mechanical deformation during breathing. This behavior was described in detail in [24]. The breathing status is usually categorized into two types: normal (eupnea) and abnormal (apnea) breathing. Abnormal respiration is a very important indicator of chronic diseases such as chronic obstructive lung disease, pneumonia or physiological instability, so the demand for a continuous respiratory system in real time with a relatively low cost is very important in practice.

The signal acquisition of the proposed system was done by a female adult in a real environment. Several precautions are considered during measurements, avoiding any movements or action by other humans. For normal breathing, the participant was asked to follow several conditions:Respiration with small movement (walking);Respiration with speaking activities (reading a book);Stable pause without any other activities.

To acquire abnormal breathing, the signal was produced by pausing respiration for a few seconds to imitate apnea, because real abnormal respiration is difficult to obtain.

Figure 4 summarizes the proposed approach to detect and classify human breathing status. In the experiment, the transmitter module was stitched into a T-shirt and was worn by the participant, as shown in Figure 5. The base station was prepared, and the receiver Bluetooth module was connected to a laptop. The base station records continuously the transmitted received signal strength indicator emitted by the embroidered antenna sensor. The participant was breathing according to the above conditions. The RSSI values were displayed in IDE software, and MATLAB read the data and plotted it in real time through the serial port. The measured raw RSSI data and related smoothed RSSI curves of eupnea breathing signals for different conditions are presented in Figure 6. The smoothed RSSI signal can be obtained by the Loess fitting method using data analysis and graphic software. The signal of normal breathing (eupnea) was generated, a signal that is similar to sinusoidal breathing, and the breathing period was clear and could be distinguished. Therefore, expiration (exhalation) and inspiration (inhalation) phases in each respiration period were clearly extracted for different examples of the normal respiration status. The experimental results demonstrated that breathing signals can be acquired wirelessly by the RSSI via Bluetooth. The RSSI range change was from −80 dBm to −72 dBm, −88 dBm to −79 dBm and −85 dBm to −80 dBm during expiration and inspiration for normal breathing, speaking and moving, respectively. The RSSI measurements were recorded for the participant at 1 m from the base station. Note that during movement, the participant maintains the physical distance between the receiver and transmitter. We can observe that the transmitted received signal strength oscillates as the participant breathes. The respiration signal is obtained with 11 breathing cycles in 60 s. Note that normal breathing rates for an adult person range from 11 to 16 breaths per minute according to the respiratory rate chart [25]. The expiration and the inspiration period are estimated to be 5 s, which is in a good agreement with the regular breathing cycle from 3 s to 6 s. An example of abnormal (apnea) breathing status is presented in Figure 7. No expiration and inspiration over a very short period imitates apnea (absence of breathing).

## 4. Discussion

A novel, fully embroidered meander dipole antenna-based sensor integrated into a cotton T-shirt for real-time respiration monitoring using the technique based on chest wall movement analysis was developed in our group. This study demonstrated the capability of the embroidered antenna sensor to detect different breathing patterns [24]. However, this breathing monitoring technique is inconvenient because the breathing mechanism uses a connecting cable for VNA that restricts the use of the proposed system. To improve the user comfort, an e-textile T-shirt with wireless communication of an embroidered antenna-sensor for real-time breathing detection was developed in this work. The proposed system is composed of a contactless antenna-based sensor integrated into a cotton T-shirtthat is connected to a transmission integrated Bluetooth module and a laptop with a receiver Bluetooth module. The area of the proposed system is 4.5 × 0.48 cm^2^, and the transmitter module is 2.36 × 3.17 cm^2^. The breathing signal is recorded through the detection of the transmitted received signal strength indicator (RSSI) by means of a base station (laptop). Table 1 presents a comprehensive comparison between two proposed systems for breathing monitoring. The proposed work demonstrated the ability of the embroidered antenna-based sensor to detect the breathing signal in real time and communicate the data via a Bluetooth protocol at 2.4 GHz to a base station. Our breathing detection systems embedded into a commercially textile T-shirt are presented in Figure 8. The respiratory antenna sensor is placed on the middle of the chest position of the human body to monitor breathing in real time by means of the stretching deformation of the textile under the movement of the chest during breathing. The working principle is based on the resonant frequency shift using the vector network analyzer (VNA) (Figure 8a). This technique makes the user uncomfortable due to the connection cable of the VNA. Therefore, we proposed an e-textile T-shirt based on the received signal strength indicator (RSSI) detected wirelessly using a base station (Figure 8b) that presents a high level of comfort for the user. The reliability of the proposed system will be addressed in future work. Several studies on the durability and washability of textile antennas have been presented in the literature [26,27]. Different coating materials (latex, silicone and so on) are used to protect the antenna from detaching in the laundry. According to those studies, the effect of coating materials on antenna performance is far less significant than manufacturing tolerances. Table 2 provides a comparison of the proposed work with other studies that monitored breathing with different methods, and we can see that a few textile sensors were found in the literature. The main novelty of our research is the development of a new wireless communication platform for breathing monitoring using a fully embroidered antenna sensor embedded into a commercial T-shirt and connected to a transmission Bluetooth module, which offers a compact size compared with reported works, provides a high level of comfort for the user and is comfortable for long-term use.

## 5. Conclusions

The feasibility of a contactless device for human breath detection based on a fully embroidered antenna-based sensor embedded into a commercial T-shirt has been presented and tested. An antenna sensor has been fully integrated into an e-textile T-shirt with wireless capability. We have demonstrated the ability of the wireless communicating portable system to monitor breathing in real time and communicate the data via a Bluetooth protocol at 2.4 GHz to a base station. The proposed wireless system demonstrated experimentally the breathing signal by means of the received signal strength indicator (RSSI) via standard Bluetooth protocol. The capability and accuracy of the proposed wearable device to detect in real time a breathing signal of a female volunteer has been demonstrated. The obtained results could make our proposed e-textile T-shirt a potential device for healthcare applications, such as in chronic obstructive pulmonary illness or sleep apnea and various respiration disorders in neonates, children and adults.

## Figures and Tables

**Figure 1 sensors-22-08667-f001:**
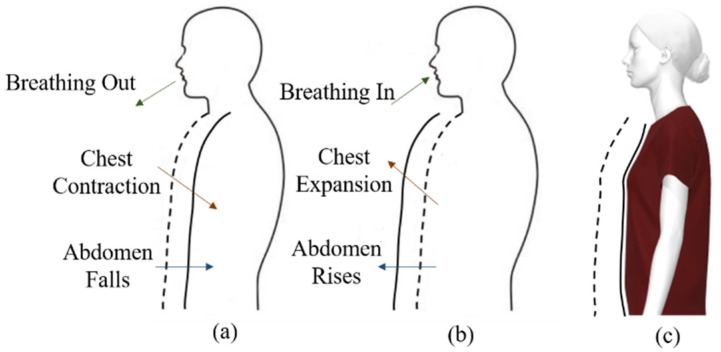
Schematic representation of the ventral body cavity: (**a**) expiration, (**b**) inspiration and (**c**) chest movement during inhalation and exhalation.

**Figure 2 sensors-22-08667-f002:**
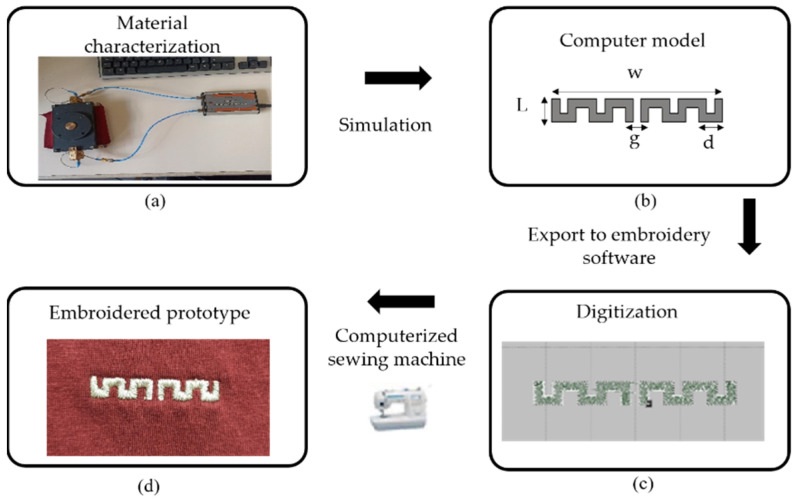
Main steps for the fabrication of the embroidered antenna-based sensor. (**a**) Measurement setup of the dielectric properties of material, (**b**) Antenna design model, (**c**) Digitization and (**d**) Embroidered prototype.

**Figure 3 sensors-22-08667-f003:**
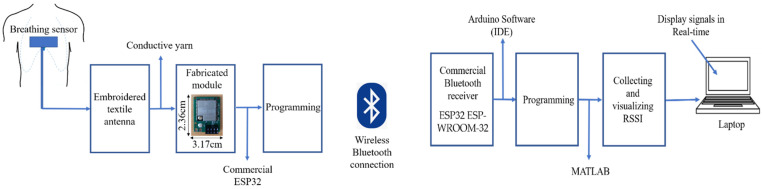
Block diagram of breathing monitoring.

**Figure 4 sensors-22-08667-f004:**
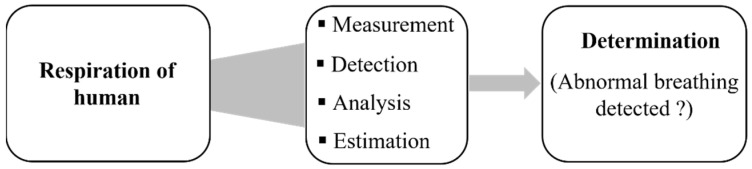
Procedure to detect human breathing status.

**Figure 5 sensors-22-08667-f005:**
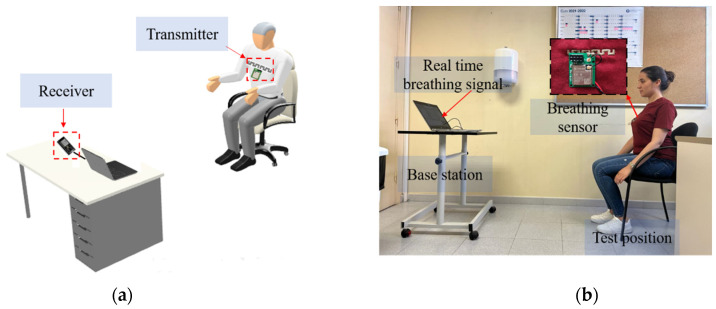
Person under test wearing the e-textile for breathing monitoring: (**a**) experimental setup configuration and (**b**) photograph of experimental setup.

**Figure 6 sensors-22-08667-f006:**
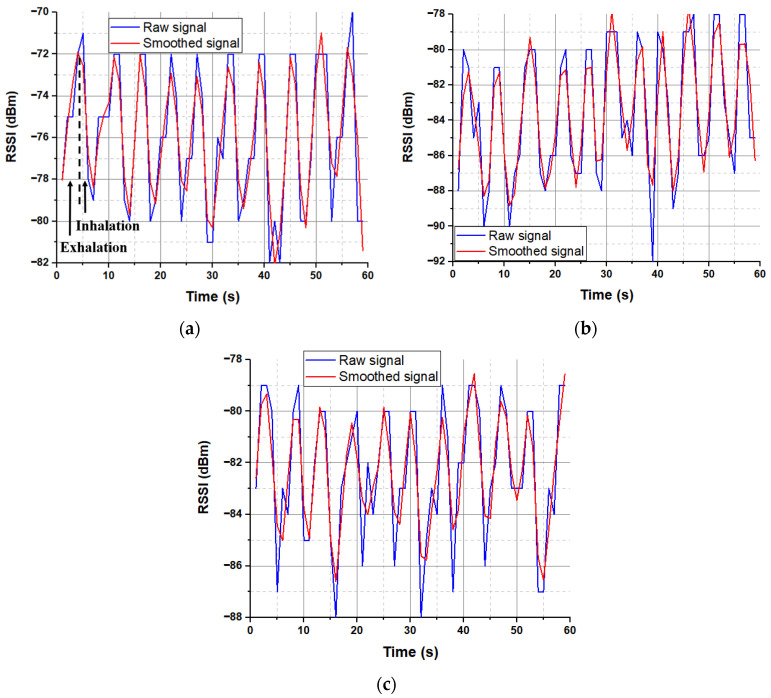
Examples of eupnea breathing signals: (**a**) normal breathing in stable position, (**b**) normal breathing while speaking and (**c**) normal breathing while moving.

**Figure 7 sensors-22-08667-f007:**
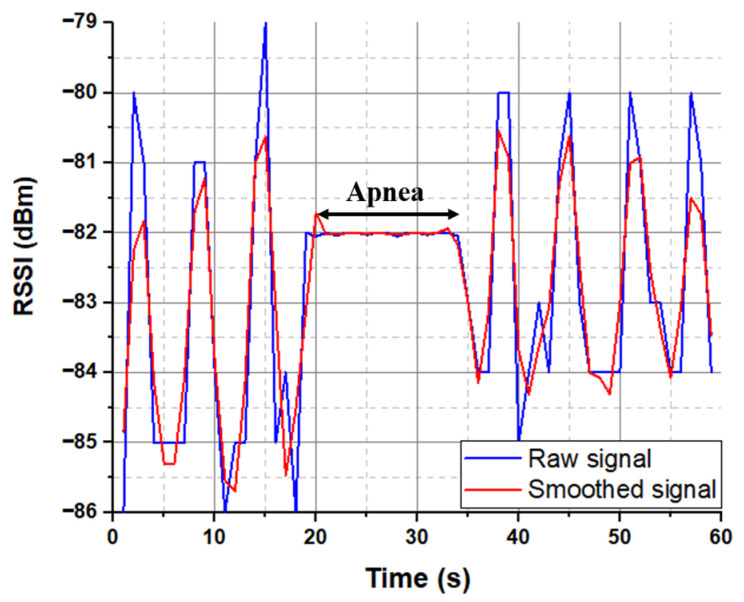
Example of abnormal breathing status.

**Figure 8 sensors-22-08667-f008:**
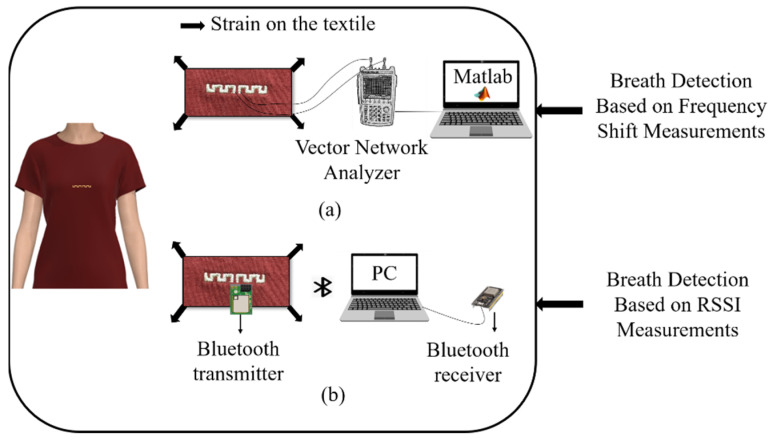
Breathing detection systems embedded into textile T-shirt: (**a**) system 1 (our previous work) based on the resonant frequency shift using VNA, (**b**) proposed system based on the RSSI detected wirelessly using a base station.

**Table 1 sensors-22-08667-t001:** Comparison between two systems for breathing monitoring.

Breathing System	Sensor Type	Measuring Parameter	Technique	Advantages	Disadvantages
System 1 (our previous work)	Meander dipole antenna sensor connected to a SMA	Frequency shift of the S_11_	Vector Network Analyzer (VNA)	Wide frequency range operation. No signal interference.	Not comfortable for long term use. Movement restriction Requires a connection cable for VNA. Not portable.
System 2 (our current work)	Meander dipole antenna sensor connected to a Bluetooth transmitter	Received Signal Strength Indicator (RSSI)	Wireless communication with a portable base station	Portable. Comfortable. User friendly. Low energy consumption.	The signal strength depends on the distance between the transmitter and the detection base station. Interference with another RF signal at 2.4 GHz.

**Table 2 sensors-22-08667-t002:** Comparison with other breathing systems.

Ref	[28]	[29]	[30]	[31]	[32]	[33]	[34]	This Work
**Sensor type**	Microphone	Thermistor	Spiral antenna	Camera	Piezoelectric	Resistive	Multimodal Patch	**Dipole antenna**
** ^1^ ** **C.T**	Mobile phone	^2^ PAT	^3^ BT	Wired	WatchPAT	wired	WIFI	** ^3^ ** **BT**
**Method**	Recording sound	Nose airflow	RSSI	Image analysis	Raw signal	^4^ RIP	Signal amplitude	**RSSI**
**Real-time**	Yes	Yes	Yes	Yes	Yes	No	Yes	**Yes**
**Textile**	No	No	Yes	No	No	Yes	No	**Yes**
**Size (mm^2^)**	450 × 250	_	200 × 100	50 × 76	70 × 60	310 × 40	65.53 × 26.67	**45 × 4.87**

^1^ C.T: Communication technology. ^2^ PAT: Portable analogue transmission. ^3^ BT: Bluetooth. ^4^ RIP: Respiratory Inductive Plethysmograph.

## Data Availability

Not applicable.

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
