# Peer review of "Wireless Communication Platform Based on an Embroidered Antenna-Sensor for Real-Time Breathing Detection"

_sensors, 2022, doi:10.3390/s22228667_

Round 1

Reviewer 1 Report

In their paper “E-Textile integration of an Embroidered Antenna-Sensor for 2 Real-time Breathing Detection” Mariam El Gharbi et al. proposed Embroidered Antenna as breathing sensing devices, a good idea and a promising application for antenna-based sensors.

The paper highlights the interest and capability of this Embroidered Antenna-Sensor for breathing detection, however some revisions should be added into,

1. the design process and observation on antenna sizes should be presented detailly.

2. the sector, ‘3.1. Mechanism of breath detection’, should be rewritten. In their original writing, more is breath mechanism and less detection mechanism.

3. Embroidered Antenna-Sensor is not a novel idea for monitoring breathing, so the novelty of the paper should be more highlighted.

Author Response

First of all, we appreciate the reviewers time and effort to provide their valuable feedback on the article. We are grateful for their insightful comments on the article. We have incorporated the changes to reflect the suggestions provided by the reviewers. We have highlighted the changes within the manuscript.

Reviewer #1:

We appreciate the comments from the reviewer and our responses are shown as follows:

In their paper “E-Textile integration of an Embroidered Antenna-Sensor for Real-time Breathing Detection” Mariam El Gharbi et al. proposed Embroidered Antenna as breathing sensing devices, a good idea and a promising application for antenna-based sensors.

The paper highlights the interest and capability of this Embroidered Antenna-Sensor for breathing detection, however some revisions should be added into.

  1. The design process and observation on antenna sizes should be presented detailly.

Response 1:

We appreciate the reviewer’s comment. For the design process, we have started with a conventional dipole antenna which has a large size, and we carried out a parametric study of different parameters in order to minimize the size and to optimize its behaviour at 2.4 GHz. Moreover, the overall size of the proposed meander dipole antenna is reduced by up to 54% compared to the normal textile dipole reported in the literature [23].

This paragraph has been included in Section 2.2 (Line 145-150).

  1. The sector, ‘3.1. Mechanism of breath detection’, should be rewritten. In their original writing, more is breath mechanism and less detection mechanism.

Response 2:

We appreciate the reviewer’s comment. The manuscript has been modified by expanding the details in Section 2.1 (Line 122-132).

The breathing sensing mechanism of the proposed system is based on the operation frequency shift of the embroidered antenna-sensor. When the textile antenna-based sensor is worn by a human body, its geometry is subject to significant deformation due to the body respiration expansion/contraction. As a result, the operation frequency of the embroidered antenna shifts downward or upward because of the strain applied on the antenna by the chest movement. The breathing antenna sensor is connected to a Bluetooth transmitter which is placed on the middle of the chest position. The detuning of the antenna frequency generates a mismatching in the transmitter which affects to the received RSSI. A base station contains a laptop with a receiver Bluetooth module prepared to continuously record the variation of the RSSI signal emitted by the embroidered antenna-sensor.

  1. Embroidered Antenna-Sensor is not a novel idea for monitoring breathing, so the novelty of the paper should be more highlighted.

Response 3:

According to the reviewer’s comment, the main novelty of our research is the development of a new wireless communication platform for breathing monitoring using a fully embroidered antenna-sensor embedded into a commercial T-shirt and connected to a transmission Bluetooth module which offers a compact size compared with reported works and provides a high comfort for the user and comfortable for long term use.

This paragraph has been included in Section 4 (Line 322-326).

Reviewer 2 Report

The manuscript by Gharbi et al. demonstrates antenna-based sensors for textile-based breathing detection. The work is interesting and technically sound. It is well written and will catch the interest of readers. Below are some points to be considered in the revision.

1. Page 2, Line 69-70 “In addition, they require complex techniques for processing data.” This part requires additional references or further clarification. Why are the mentioned techniques more complex than the approach used in this study?

2. How do authors deal with noise associated with any other human actions/movements/activities and false body signals? Which signal processing strategies were implemented or which assumptions were made? They should be clearly mentioned in the manuscript.

3. It is also important to describe the textile part in detail. Should textile be a compression garment? In other words, how do authors manage firm attachment of sensors to the chest part? Do authors use tape, etc.?

4. In this study authors used two different wireless electronic systems: antenna and Bluetooth. Would it not be enough to design this system only with antenna-based wireless sensors? Because Bluetooth requires a battery and has additional technical challenges and requirements.

5. In the long-term, coating of nylon threads with chemicals will raise the questions about health concerns and reliability of the system. This will also challenge the washability of these textiles. What are the possible ways to handle this topic? Would be great to discuss this in the discussion.

6. Figures are not in desired paper quality. For example, Figure 2 subfigures are too small. Figure 3 does not contain too much information (I would suggest adding more data or info from the study). Figure 6&7 sub figures are too big.

7. It is important including state of the art studies in flexible and textile-based physiological health monitoring and computational platforms. Particularly, recent developments in fiber and textile-based technology would be suitable to mention. For example, below approaches provide unique physiology monitoring solutions which were not covered in the manuscript.

a) Rein, M., et al. Diode fibres for fabric-based optical communications. Nature 560, 214–218 (2018).

b) Dagdeviren, C., et al., Conformal piezoelectric energy harvesting and storage from motions of the heart, lung, and diaphragm. PNAS, 111, 1927 (2014).

Reviewer 3 Report

This paper mainly studies a wireless breath detection method based on textile antenna-sensor. The sensor is integrated into the e-textile T-shirt with Bluetooth transmitter to detect breath in real time and wirelessly, which solves the problems of poor comfort and user restriction of the existing wired detection system. The proposed e-textile T-shirt has a good application prospect in the diagnosis of chronic obstructive pulmonary disease or sleep apnea and various respiratory diseases in neonates, children and adults.

There are some problems, which must be solved before it is considered for publication. It will be better if the following problems are well-addressed.

1. At the end of the first paragraph, it is said that the wireless system has long-term usability. It is suggested to explain why it has long-term usability or provide references or basis.

2. The title of the second chapter should be 2.1, 2.2, 2.3 .

3. The first part of the second chapter, "Mechanism of breath detection", mainly describes the mechanism of respiration rather than the mechanism of breath detection. The respiratory detection mechanism should mainly explain what principle or method is used to detect respiratory signals (i.e. how to convert chest and abdomen movements generated by respiration into electrical signals to be measured)

4. Is it more appropriate to replace the subtitle "Respiratory antenna-sensor based on RSSI measurement" with "Respiratory sensor system based on RSSI measurement"? This section mainly describes the whole breath detection system, including antenna sensor, transmission Bluetooth module, receiver Bluetooth module and base station.

5. It is suggested to analyze the causes of changes in RSSI caused by inspiratory and expiratory strains according to respiratory test results (RSSI data). That is, analyze whether the signal strength becomes stronger or weaker due to sensor stretching, and explain why.

6. The upper and lower contents in Table 2 are repeated and need to be modified.

Author Response

First of all, we appreciate the reviewers time and effort to provide their valuable feedback on the article. We are grateful for their insightful comments on the article. We have incorporated the changes to reflect the suggestions provided by the reviewers. We have highlighted the changes within the manuscript.

Reviewer #3:

We appreciate the comments from the reviewer and our responses are shown as follows:

This paper mainly studies a wireless breath detection method based on textile antenna-sensor. The sensor is integrated into the e-textile T-shirt with Bluetooth transmitter to detect breath in real time and wirelessly, which solves the problems of poor comfort and user restriction of the existing wired detection system. The proposed e-textile T-shirt has a good application prospect in the diagnosis of chronic obstructive pulmonary disease or sleep apnea and various respiratory diseases in neonates, children and adults.

There are some problems, which must be solved before it is considered for publication. It will be better if the following problems are well-addressed.

  1. At the end of the first paragraph, it is said that the wireless system has long-term usability. It is suggested to explain why it has long-term usability or provide references or basis.

Response 1:

The wireless system has long-term usability, because it can be used for patients to monitor continuously their breathing activity patterns and that could provide additional understanding about disease progression, allowing prompt clinical intervention.

According to the reviewer’s comments, some references were added in the manuscript [6,7].

This paragraph has been included in Section 1 (Line 49-52).

“Because they allow a long-term usability providing convenient continuous monitoring of vital signs and that could provide additional understanding about disease progression, allowing prompt clinical intervention because the data is sent and received wirelessly in real time [6,7].”

  1. The title of the second chapter should be 2.1, 2.2, 2.3.

Response 2:

We appreciate the reviewer’s comment. The title numbers have been updated in the revised manuscript.

  1. The first part of the second chapter, "Mechanism of breath detection", mainly describes the mechanism of respiration rather than the mechanism of breath detection. The respiratory detection mechanism should mainly explain what principle or method is used to detect respiratory signals (i.e. how to convert chest and abdomen movements generated by respiration into electrical signals to be measured).

Response 3:

We appreciate the reviewer’s comment. The manuscript has been modified by expanding the details in Section 2.1. (Line 122-132).

The breathing sensing mechanism of the proposed system is based on the operation frequency shift of the embroidered antenna-sensor. When the textile antenna-based sensor is worn by a human body, its geometry is subject to significant deformation due to the body respiration expansion/contraction. As a result, the operation frequency of the embroidered antenna shifts downward or upward because of the strain applied on the antenna by the chest movement. The breathing antenna sensor is connected to a Bluetooth transmitter which is placed on the middle of the chest position. The detuning of the antenna frequency generates a mismatching in the transmitter which affects to the received RSSI. A base station contains a laptop with a receiver Bluetooth module prepared to continuously record the variation of the RSSI signal emitted by the embroidered antenna-sensor.

  1. Is it more appropriate to replace the subtitle "Respiratory antenna-sensor based on RSSI measurement" with "Respiratory sensor system based on RSSI measurement"? This section mainly describes the whole breath detection system, including antenna sensor, transmission Bluetooth module, receiver Bluetooth module and base station.

Response 4:

According to the reviewer’s comments, the subtitle has been updated in the manuscript.

  1. It is suggested to analyze the causes of changes in RSSI caused by inspiratory and expiratory strains according to respiratory test results (RSSI data). That is, analyze whether the signal strength becomes stronger or weaker due to sensor stretching, and explain why.

Response 5:

The antenna-based sensor is sewn into the T-shirt at chest position where it can sense the chest contracting and expanding with each breath. The operation frequency of the embroidered antenna shifts downward or upward because of the strain applied on the antenna by the chest movement. The breathing antenna sensor is connected to a Bluetooth transmitter which is placed on the middle of the chest position. The detuning of the antenna frequency generates a mismatching in the transmitter which affects to the received RSSI.  

  1. The upper and lower contents in Table 2 are repeated and need to be modified.

Response 6:

According to the reviewer’s comments, the Table 2 has been modified in the manuscript.